# Algorithmic Probability Method Versus Kolmogorov Complexity with No-Threshold Encoding Scheme for Short Time Series: An Analysis of Day-To-Day Hourly Solar Radiation Time Series over Tropical Western Indian Ocean

**DOI:** 10.3390/e21060552

**Published:** 2019-05-31

**Authors:** Miloud Bessafi, Dragutin T. Mihailović, Peng Li, Anja Mihailović, Jean-Pierre Chabriat

**Affiliations:** 1Faculty of Sciences and Technology, University of La Réunion, Laboratoire d’Energétique, d’Electronique et Procédés, 15 Avenue René Cassin, Sainte-Clotilde, 97715 La Réunion, France; 2Faculty of Agriculture, University of Novi Sad, Dositej Obradovic Sq. 8, 21000 Novi Sad, Serbia; 3ACIMSI—Center for Meteorology and Environmental Modeling, University of Novi Sad, 21000 Novi Sad, Serbia

**Keywords:** hourly solar radiation, western Indian Ocean, tropical meteorology, complexity measures, algorithmic probability method, coding theorem, Kolmogorov complexity, binary encoding

## Abstract

The complexity of solar radiation fluctuations received on the ground is nowadays of great interest for solar resource in the context of climate change and sustainable development. Over tropical maritime area, there are small inhabited islands for which the prediction of the solar resource at the daily and infra-daily time scales are important to optimize their solar energy systems. Recently, studies show that the theory of the information is a promising way to measure the solar radiation intermittency. Kolmogorov complexity (KC) is a useful tool to address the question of predictability. Nevertheless, this method is inaccurate for small time series size. To overcome this drawback, a new encoding scheme is suggested for converting hourly solar radiation time series values into a binary string for calculation of Kolmogorov complexity (KC-ES). To assess this new approach, we tested this method using the 2004–2006 satellite hourly solar data for the western part of the Indian Ocean. The results were compared with the algorithmic probability (AP) method which is used as the benchmark method to compute the complexity for short string. These two methods are a new approach to compute the complexity of short solar radiation time series. We show that KC-ES and AP methods give comparable results which are in agreement with the physical variability of solar radiation. During the 2004–2006 period, an important interannual SST (sea surface temperature) anomaly over the south of Mozambique Channel encounters in 2005, a strong MJO (Madden–Julian oscillation) took place in May 2005 over the equatorial Indian Ocean, and nine tropical cyclones crossed the western part of the Indian Ocean in 2004–2005 and 2005–2006 austral summer. We have computed KC-ES of the solar radiation time series for these three events. The results show that the Kolmogorov complexity with suggested encoding scheme (KC-ES) gives competitive measure of complexity in regard to the AP method also known as Solomonoff probability.

## 1. Introduction

Southern tropical hemisphere is mainly covered by the ocean. As its northern counterpart, this is an area on the earth where there is a high amount of solar resource. The tropical western Indian Ocean (40°E–65°E/30°S–15°N) is an area over which there is numerous inhabited islands (Madagascar, Mauritius, Rodrigues, La Réunion, Mayotte, Comores, Seychelles, and La Réunion). These islands cover a surface of 594,592 km² with a total of 30,236,310 inhabitants. Thus, in the context of climate change, solar resource is an opportunity for these disseminated islands in this part of the world for improving their sustainable development and a hopeful social and economic capacity building. Despite this high potential for solar radiation resource, the energy received on the ground is exceedingly variable in space and in time over the ocean. Following the bulk sun path through the year, the primary time scale variability of solar radiation prevailing over the tropical ocean region is within a daytime range. Nevertheless, within this time scale, the solar radiation fluctuations are driven by numerous physical factors. Among them, the radiative properties of water vapor and aerosol play an important role on the solar radiation attenuation through the atmosphere. Moreover, complex and intricate interactions with clouds have a direct impact on the reflection, transmission, and attenuation of sun light and thus affect the solar energy budget received on the ground. In addition, spatial and time cloud cover patterns are triggered by marine boundary layer turbulence and large-scale synoptic circulation in the tropical troposphere. Thus, the space and time variability of daily solar radiation is very complex and its forecast is still a challenge for solar resource research [1,2]. As mentioned previously, the predictability of solar radiation remains a social and economic stake for small islands located in the tropical ocean area. Since last ten years, information theory and computing science can contribute to tackle this challenge through new methods including data mining and algorithmic complexity. For the latter, Kolmogorov complexity (hereafter KC) does not refer only to scientific studies on computing science. It is also related to fields like medicine, psychology, biology, physics, solar energy, and many others [3,4,5]. This means that the theory of the information has given a new insight and has a significant contribution to measure the real-world complexity [6]. Usually, complexity contained in one time series is often related to the physical concept of entropy. However, based on Turing’s machine and universal lossless data compression, KC gives another framework of meaning which is how minimal computational information is contained in one time series or a set of strings. Such approach can be considered as a measure of the degree of regularity or irregularity in a set of binary numbers and how much the information is compressible. Commonly, Lempel-Ziv’s algorithm is a widely-used universal lossless data compression tool to achieve this goal [7]. Recent studies on daily solar radiation emphasize that the use of such algorithmic method (through the Kolmogorov time that is inverse to KC) remains a reliable tool for estimating solar resource predictability over small tropical islands [8,9]. Nevertheless, there is a weakness about predictability with Kolmogorov time on solar radiation because of the drawback in encoding procedure of time series for calculation of KC. With this method, we have to preprocess the physical time series and transform it into a binary data set; but to achieve that, we need to define a threshold value to assigned binary values for each value in the time series in regard to its value exceeding or not exceeding the selected threshold value. Thus, binarizing the time series with this method is sensitive to the threshold value. The Kolmogorov complexity (KC) as a measure does not distinguish between time series with different amplitude variations and similar random components. Note that when we convert a time series into a string, then its complexity is hidden in the coding rules; namely, in the procedure of establishing a threshold for coding, some information about the structure of the time series can be lost. To overcome this problem, Mihailovic et al. [10] introduced the Kolmogorov complexity spectrum to assess how the computation of KC is sensitive to the threshold value used to encode the time series. Bessafi et al. [9] showed that, for 5-years daily solar radiation time series, the mean is an optimum threshold. Furthermore, they mentioned the ability of KC computation to give an estimation of daily solar radiation predictability but emphasizing the limitation of this method for shorter time series. To embed the purpose of this study, it is pointed out that the two previous studies [8,9] focused on complexity of solar radiation at La Réunion with spatially sparse daily dataset. The former study was achieved using 11 laboratory solar radiation stations throughout the 2013–2015 period to assess the spatial complexity pattern over the island. A weighted Hamming distance was introduced to smooth the geographical complexity and overcome the sparsity of the data. The results reveal that the spatial complexity pattern is well-related to the spatial local meteorological weather pattern prevailing at La Réunion (trade winds that circulate in a “flow around” regime). The latter study used higher dense, but still sparse, daily solar radiation dataset using 32 stations of Météo-France (French meteorological office) available for the 2011–2015 period. This study showed that the spatial complexity pattern over La Réunion is also variable in time and sensitive to large-scale climate events (2011–2012 La Nina event and preceding a very strong El-Nino 2015–2016). Following these two studies, the question is how the complexity looks like spatially and temporally for a larger area at infra-day time scale. Solar radiation satellite data can contribute to give an answer. Moreover, such data are commonly spatially available at regular grid with infra-day time sampling as well. However, the first task is: If we would like to extend those previous predictability studies on daily solar radiation to infra-day sampling data with KC computation used in the previous studies, we should have in mind that it is inapplicable for hourly solar data time series. It means that we have to overcome the latter limitation of KC method for hourly solar radiation by another method which is detailed after. Studies on hourly solar radiation complexity is also of an important interest to make solar energy and its exploitation more efficient and economically acceptable. Therefore, we still focus on Kolmogorov complexity using Lempel-Ziv algorithm but for the hourly solar radiation observed over a larger geographical area embedding La Réunion. Nevertheless, to overcome the limitation of KC of computation for daytime hourly data, we have inserted a new encoding scheme instead of the threshold one used in the original KC method (threshold scheme to binarize data plus Lempel-Ziv compression with normalization) to be able to have a measure of complexity of short string. Our approach can namely be considered as an extension of KC with a new encoding scheme (hereafter KC-ES) without threshold process.

To assess the reliability of this extended KC method (encoding without threshold plus Lempel-Ziv compression with normalization), we have compared our method with another method based on an algorithmic and probabilistic approach. Suggested by Delahaye et al. and Soler-Toscano et al. [11,12], coding theorem and the algorithmic probability (hereafter AP) is able to compute complexity for short time series. The AP method is used here as a benchmark method to evaluate the KC-ES method.

The purpose of this study is to compute the complexity of half-day hourly solar radiation dataset from satellite observations over the western Indian Ocean during the 2004–2006 time period using two information measures, which are coding theorem (hereafter CT) with a namely algorithmic probability and Kolmogorov complexity with new encoding scheme (KC-ES) methods. The study is organized as follows. Section 2 describes (i) the coding theorem (CT) and AP method; (ii) the KC method; (iii) and the encoding scheme expanding hourly solar radiation into longer binary string (KC-ES). Section 3 presents the studied area, the solar variability over the tropical western Indian Ocean and provides information about the main meteorological patterns that prevailed during the 2004–2006 period. Finally, Section 4 includes both presentation of the results obtained and discussion. The concluding remarks are given in Section 5.

## 2. Material and Methodology

As mentioned previously, solar radiation received on the ground over the tropical belt is higher than the other latitudinal bands as extratropical and polar latitudes. The variability of solar radiation is dependent on the weather conditions at various space and time scale range. For the latter, it is challenging to catch the infra-day nonlinear and nonstationary behavior of solar radiation time series with one-hour time sampling. To study the day-to-day variability of infra-day solar radiation, first, we briefly describe the coding theorem for calculating the complexity of one-hour time sampling time series which can be considered as short ones. Additionally, we present calculation of algorithmic complexity of a binary time series obtained by using (i) a threshold value in encoding scheme and (ii) a new encoding scheme we suggested to overcome the drawback of KC measure for short-length binary set.

### 2.1. Coding Theorem (CT) and Algorithmic Probability (AP) Method 

Major drawback using KC method is an inaccuracy in its application for short binary strings [11,12]. This can be demonstrated by the invariance theorem which states that if $U_{1}$ and $U_{2}$ are two universal Turing machines and $K_{U_{1}}\left(S\right)$ and $K_{U_{2}}\left(S\right)$ are the algorithmic complexity (KC) of a binary string S for $U_{1}$ and $U_{2}$, then there exists a constant c that does not depend on S such that for all binary string $\left|K_{U_{1}}\left(S\right)-K_{U_{2}}\left(S\right)\right|<c$. An algorithmic complexity (KC) of a binary string S is defined as the shortest program p with which the output of a universal Turing machine U is S. Formally, we can write this definition as $K_{U}\left(S\right)=min_{p}\left\{\left|p\right|:U\left(p\right)=S\right\}$. In practice, c can be arbitrarily large and gives less accurate KC measure for short strings. To overcome the limitation about short binary strings, the coding theorem (CT) sets a relationship between KC and AP method [13,14]. As mentioned by Levin, the algorithmic probability (AP) of a string S is defined as the probability that a random program would give S with a Turing machine U. The coding theorem (CT) states that for any string S and prefix-free universal Turing machine U, $\left|-log_{2}\left(m\left(S\right)\right)-K_{U}\left(S\right)\right|\leq M$, for a fixed number M not dependent of S. m(S) is the Levin’s semimeasure or the universal distribution (0<m(S)<1). This algorithmic probability for a program p is defined as $m\left(S\right)=\underset{\left\{p:U\left(p\right)=S\right\}}{\sum }1/2^{\left|p\right|}$ where |p| is the length of the program p that produces S [13]. In fact, it is well-known that it is impossible to state if a Turing machine halts on or not to produce an arbitrary string S with a program p [13,14,15]. Usually, to compute KC of a binary string S from its frequency, we use the approximate formula$\;K\left(S\right)\approx -log_{2}m\left(S\right)$ [16,17,18]. To obtain the probability distribution for a string S, we need to run a huge number of randomly selected Turing machines. However, this computation is very challenging theoretically and numerically. In practice, for this study, we have used freely available probability distribution to compute the complexity of short binary strings [13]. There are available numerical tables with values of the algorithmic complexity dedicated for short strings. For more details on this subject, the reader either can find in Reference [13] or consult an online algorithmic complexity calculator (OACC) following the website link http://www.complexitycalculator.com.

### 2.2. Kolmogorov Complexity (KC)

To compute KC of the hourly solar radiation time series x(t), we need firstly to encode the time series before using the Lempel-Ziv algorithm [7] in order to replace it by a new set of binary values such as:(1)$$\left(t\right)=\{\begin{array}{c} 0\;\;\;\;if\;x\left(t\right)<x^{Threshold}\\ or\;\;\;\;\;\;\;\;\;\;\;\;\;\;\;\;\;\;\;\;\;\;\;\;\;\;\;\;\;\;\;\;\;\;\;\;\;\;\;\;\\ 1\;\;\;if\;\;x\left(t\right)\geq x^{Threshold} \end{array}$$
S where$\;x^{Threshold}$ is a threshold value [8,9]. Usually, the mean of the time series x(t) is used as a threshold. Following this preprocessing step, we search in the binary time series S(t) the overall possible subset sequences which are different from each other. The number of nonmatching subsets represents the complexity of the series. Therefore, the value of C(N) involved in the binary template S(t) is increasingly proportional to randomness. Asymptotically, when the length N of the binary series tends to infinity, the number C(N) tends to reach its limit, i.e., $b\left(N\right)=N/log_{2}N$. Usually, the normalization of the Kolmogorov complexity $KC_{n}\left(N\right)$ is given by $KC_{n}\left(N\right)=C\left(N\right)/b\left(N\right)=C\left(N\right)log_{2}N/N$.

### 2.3. Encoding Hourly Solar Radiation and Kolmogorov Complexity (KC-ES)

Day-to-day hourly solar irradiation value over tropical area is expected to be greater than zero only during half-days which means that the number of values we can expect to have is between eleven and twelve depending the seasonal variation of daylength. As mentioned previously, KC is not accurate for short binary strings [11,12]. To overcome this, we propose a new approach which is to encode each hourly value by its binary representation. Let us assume that for one daylight, we have M values of hourly solar radiation time series $x\left(t\right)=\left\{x\left(t_{1}\right),\;x\left(t_{2}\right),\;\ldots ,\;x\left(t_{M}\right)\right\}\;$ corresponding to the measurements of the solar radiation recorded successively hour-by-hour from time $t_{1}$ to time $t_{M}$ of the daytime. We suppose that each value of hourly solar radiation $x{\left(t_{i}\right)}_{i=1,..,M}$ is encoded to its binary representation in base 2. For example, let us assume that the value of hourly solar radiation over the tropic during midday of summer period is around 1200 W/m². Thus, the binary string 10010110000 is the binary representation with 11 digits of the integer 1200 in base 2. In addition, let us consider now we have M values of hourly solar radiation $x\left(t\right)=\left\{x\left(t_{1}\right),\;x\left(t_{2}\right),\;\ldots ,\;x\left(t_{M}\right)\right\}$ for one half-day. We can convert each integer value $x{\left(t_{i}\right)}_{i=1,..,M}$ to a binary string $s{\left(t_{i}\right)}_{i=1,..,M}$ of length of $N_{i}$. This process allows us to compute KC not with M integers but with M string $S=s\left(t_{1}\right)s\left(t_{2}\right)..s\left(t_{M-1}\right)s\left(t_{M}\right)$ of length $N_{i}{}_{\left\{i=1,\;\ldots ,\;M\right\}}$ each one. Then, the concatenation of M binary string $s{\left(t_{i}\right)}_{i=1,..,M}$ is a string of length $\left|S\right|=\overset{M}{\underset{i=1}{\sum }}N_{i}$. For example, let us assume we have M=12, that means we have half-day of hourly measured solar radiation. For each hour, we code the solar radiation integer value to 11 digits of binary string. Thus, we have converted a set of 12 values into a binary string of length |S|=132. For more clarity, let us now consider the following set of the hourly global horizontal irradiance (GHI) recorded at 30°S–57°E on 1 February 2004. These are integer values {83|515|758|906|910|857|909|887|733|526|219|25}. The vertical bar between integer values is used here as a symbolic separator to explain how we conceptually proceed to encode the time series set. Each value can be coded into its binary representation to build the set of the twelve binary strings. This set of binary strings is conceptually a juxtaposition of twelve separated strings by a symbolic separator (vertical bar) as it is symbolically shown in the following way {1010011|1000000011|1011110110|1110001010|1110001110|1101011001|1110001101|1101110111|1011011101|1000001110|11011011|11001|}. Formally, the concatenation of the encoded values of hourly solar radiation into one binary string is significantly longer than the original integer’s values set. In this example, the length of the final binary string is of length 110 and represented by a long string which contains a set of successive of binary value without symbolic separator and bracket 10100111000000011101111011011100010101110001110110101100111100011011101110111101101110110000011101101101111001. The total length is simply the sum of the twelve individual binary strings having the following lengths {7|10|10|10|10|10|10|10|10|10|8|5}, respectively. This process has the advantage to expand a short set of numerical numbers to a longer binary string with which we can compute KC-ES. In addition, there is no need to proceed to the use of an arbitrary threshold to build the binarized data set as obtained with the original Kolmogorov complexity (KC) calculation with Lempel-Ziv algorithm. In comparison, the binarization using the threshold method gives the binary string 001111111000, whereas the new encoding method gives the binary string 10100111000000011101111011011100010101110001110110101100111100011011101110111101101110110000011101101101111001. In resume, this encoding method is achieved to (i) extend the length of the string using a common binary conversion to overcome the limitation of Kolmogorov complexity computation to short binary strings and (ii) build a binarized data set without threshold process. 

## 3. Solar Variability and Tropical Meteorological Weather over West Indian Ocean

Our interest of the western part of the Indian Ocean is that there are inhabited islands which are Madagascar, Mayotte, Comores, Mayotte, Seychelles, and Mascarenes Islands (Mauritius, Rodrigues, and La Réunion) for which solar energy resource is important. The geographical locations of these islands cover the south tropical band (0°-30°S) and the 43°E–62°E longitude band. In this study, we used hourly GHI (global horizontal irradiance) measurements recorded during the 2004–2006 period and freely available at the satellite-derived HC3 archives from Solar Radiation Data (http://www.soda-pro.com). The spatial coverage of the data is 66°S–66°N/66°W-66°E. To evaluate the complexity with AP method and KC-ES method of hourly solar radiation time series over the western part of the Indian Ocean, we have selected four longitude transect (43°E, 52°E, 57°E, and 62°E) from 30°S to 15°N with a latitudinal increment of one degree. As shown in Figure 1, all these transects are representative of tropical ocean area except for the 43°E longitude transect which crosses also the land of the horn of Africa (Kenya, Somalia, and Ethiopia).

As mentioned previously, solar radiation is higher in the tropics and decrease towards the pole. The variability of solar radiation is latitudinal-dependent but can also vary from the west to the east of the Indian Ocean basin. As it is shown in Figure 2, there is a south-poleward increasing of the daily variability of hourly solar radiation time series except for the Mozambique Channel and over the Horn of Africa land. We use the coefficient of variation (CV) as a measure of the solar radiation variability by comparing the amplitude of the fluctuations to the mean. Globally, over the equatorial region (approximately located between 10°S and 10°N) and the Indian Ocean (longitude 52°E, 57°E, and 62°E), there is a slight eastward increase of the variability of solar radiation with a CV between 0.2 and 0.4. The more pronounced variability of CV within the equatorial band (15°S–15°N) could be related to the equatorial convective activity induced by the latitudinal migration of intertropical convergence zone (ITCZ) through the year, the convectively coupled equatorial waves (Kelvin, ER for equatorial Rossby, MJO for Madden–Julian oscillation) which are frequently present over this area [19,20]. Thus, the day-to-day variability of hourly solar radiation depends on the large-scale tropical and equatorial weather conditions prevailing over the western tropical region of the Indian Ocean [21].

Figure 3 displays an example of the time solar radiation variability at 9:00 UTC observed during the 2004–2006 period. We have depicted this variability for three latitudinal bands which represent the southern tropic (30°S–15°S), the southern equatorial (15°S–0°), and northern equatorial band (0°-15N) at different longitude (43°E, 52°E, 57°E, and 62°E). It is highlighted that the more pronounced seasonal variability is for the southern tropic, whereas over the equatorial band, the solar radiation exhibits an additional semiannual cycle [22]. The average solar radiation amplitude throughout the year at midday was around 400 W/m² over the southern tropical band (30°S–15°S) and 200 W/m² over the equatorial band (15°S–15°N). Superimposed to the annual and semiannual cycle, there is also fluctuations of solar radiation at the intraseasonal cycle with some lower peaks of solar radiation which could be related to isolated cloudy days [9,22,23].

During the daytime, the hourly solar radiation over the ocean is highly changeable during the day. According to the rotation of the Earth around the sun during the year, this diurnal variability is also linked to the weather conditions and convective activities associated with large-scale mesoscale convective systems and tropical perturbations. Figure 4 shows that over the western part of the Indian Ocean, the solar radiation is on average high around 900 W/m² around midday with a maximum around 1200 W/m². We can also notice that the solar radiation variability is around 192 W/m² during midday and 73 W/m² during in the morning and the afternoon.

Before we analyze AP and KC-ES method with three case studies, let us briefly set out the meteorological patterns which trigger the cloud covers and then the solar radiation variability over the western Indian Ocean. The main tropical systems encountered in this region which drive the cloud patterns are the ITCZ (intertropical convergence zone), the trade winds over Mascarenes, the monsoon fluxes over North of Madagascar, and the large-scale tropical air–sea perturbations as ENSO (El Nino–La Nina southern oscillation), MJO (Madden–Julian oscillation), IOD (Indian Ocean dipole), and tropical cyclones [24,25,26]. From day to day, low and high clouds associated with convective systems are frequent over this area [27,28,29]. 

As mentioned previously, solar radiation received on the ground is mainly influenced by the cloud cover patterns which are often related to convective activity. Over the western Indian ocean, there are numerous clouds types from marine boundary cloud (cumulus, stratocumulus) to deep cloud extended to the whole troposphere [27,28,29]. The former is generally triggered by local weather condition (temperature, wind inversion, orographic, and thermal forcing), latent heat release from sea–air interaction, and the latter is often related to vertical instability of moist tropical atmosphere through cumulonimbus cloud type. In terms of convective activity, 2004–2006 period was an active period. During this period, we have noticed three strong meteorological events. Firstly, there were a sequence of positive episode of South-West Indian Ocean (SWIO) sea surface temperature (SST) anomalies which has significantly disturbed the thermal pattern and the atmosphere circulation of the South-West Indian Ocean. A strong MJO was detected during April 2005. Lastly, there were also nine tropical cyclones which crossed the western part of the basin during this period. Tropical cyclone genesis and tracks over the Indian Ocean are mostly linked to the state of large-scale dynamic and thermodynamic conditions of moist atmospheric air and the state of convectively-coupled equatorial waves [30]. All these meteorological perturbations have an impact on the space and time clouds pattern over the ocean and consequently affect the variability of solar radiation within various time scales [23].

To illustrate the large-scale convective activity, which mainly drives the cloud cover pattern and then the amount of solar radiation received on ground, Figure 5 displays the latitude-time and longitude-time OLR anomalies (outgoing longwave radiation). The OLR is commonly used as a proxy for deep convective activity and an OLR value less than around 200 W/m² is usually assigned to deep convection [31]. During 2004–2006, the OLR was in average 250 W/m² which is indicative that the western part of the Indian Ocean was on average an area with a persistent convective activity. In addition, we can notice that the ORL variability was around 20 W/m² over the west part of the Indian Ocean during this period. 

Anomalies mean that we have extracted the daily long-term mean from the original data. Negative or positive is related to enhanced or suppressed convective activity in regard to the mean state. As mentioned, convective activity is highly variable over the equatorial band (15°S–15°N) which increases eastward. There is a superimposed seasonal and intraseasonal variability which is mainly due in this latitude band by the ITCZ migration, activity, and equatorial convective systems coupled by wave propagation (Kelvin, ER, and MJO). Conversely, over the south tropical (30°S–15°S), the convective activity is rather driven by the seasonal cycle and intraseasonal cycle induced by extratropical or tropical connection through the extratropical Rossby waves (Figure 5a). We can also notice that there was the strongest convective activity over the equatorial band for the 52°E–62°E longitude band during the beginning of 2004 and the end of 2006. There are noticeable negative OLR anomalies less than −50 W/m² which is roughly two times greater than the standard deviation of 20 W/m² and indicative of deep convection. The longitude-time OLR known as Hovmöller diagram is shown in Figure 5b. The convective activity was more pronounced during 2004 and 2006 austral summer (November to May) over the western part of the Indian Ocean. The seasonal and intraseasonal variability of the OLR anomalies are also noticeable in this figure revealing that the studied area was triggered by convective activity which had an impact on solar radiation variability from daily to infra-daily time scale.

## 4. Results and Discussion

As it is recently shown, KC is a useful information measure to analyze the complexity of daily solar radiation time series over the South-West Indian Ocean location [8,9]. However, this measure could be inaccurate for short length binary strings. Therefore, then we have to have in mind this limitation when this complexity is calculated for hourly solar radiation time series. To overcome this drawback, we used the coding theorem (CT) and the AP method to estimate the complexity of hourly solar radiation time series. We have also taken into account the Kolmogorov complexity using a new encoding scheme (KC-ES), which overcomes the limitation of the string length and prevents the use of the threshold method in binarizing the originally-measured time series. The spatial and time of hourly solar radiation complexity during the 2004–2006 period was investigated with these two methods to establish the reliability of the use of the Kolmogorov complexity with the suggested encoding scheme for short binary strings and its capability to be used in analysis of natural time series [32,33]. The performance of KC-ES and AP methods were both estimated and compared during extreme ocean and atmospheric events which have a strong impact on the weather, the cloud cover, and then the solar radiation variability received on the ground. Firstly, ocean events occurred during the 2004–2006 period and especially with stronger intensity at the end of 2004 and the beginning of 2005. For these two latter, there was a noticeable increase of the SST over the south of Mozambique Channel. This kind of ocean event is usually encountered over the Indian Ocean and it is an interannual tropical modes like the ENSO (El Nino–La Nina southern oscillation) over the Pacific. Secondly, there were a strong MJO episode of April 2005 which was an eastward tropical atmospheric wave. This atmospheric oscillation mostly disturbed the convective activity and the cloud movements over the equatorial band (15°S–15°N). Thirdly, western Indian Ocean have experienced nine tropical cyclones which footprint the convective activity and cloud cover pattern along their track road.

### 4.1. Hourly Solar Variability and Algorithmic Probability Method

Hourly clear-sky index is used here to evaluate the atmospheric attenuation due to clouds and their impact on the stochasticity of the fluctuations [34]. Figure 6 shows the distribution of the time increment of solar radiation τ (τ = 1, 2, …, 6 h). This is displayed using a semilogarithmic scale (along the vertical axis). The departure from the Gaussian distribution is an indicator of the intermittency [9]. In this figure, the non-Gaussianity of the distribution of clear sky index $k_{T}=\frac{GHI_{observed}}{GHI_{clear\;sky}}$ with a pronounced deviation for large increments is clear for hourly solar radiation over the western part of the Indian Ocean.

In addition, beyond the intermittency of hourly clear sky index, the questions are “how complex is the variability of solar radiation during daytime ?” and “how it could be caught by the AP method ?” Over the tropical band 30°S–15°N, the length of the daytime is on average of 12 h. This means that, for hourly time sampling, the dataset length per day is short. Thus, computing KC is not possible for time series consisting of only twelve values. In this case, the use of the coding theorem (CT) is useful to compute complexity using the concept of AP method [13,35].

As mentioned previously, complexity of a binary string is deduced from its probability distribution produced by the huge number of Turing machines run [12,33]. Before computing the frequency distribution, we have binarized for each day the hourly clear sky index by using the mean as a threshold value [8,9]. Figure 7 depicts dependence of hourly solar radiation complexity on latitude. South to 5°S, the complexity is roughly constant for the 52°E–62°E longitude band. Mozambique Channel (15°S–25°S) exhibits a lower complexity which could be related to regularity of the weather as a result of the splitting flow around Madagascar and quasi-permanent trough in the Mozambique Channel [36]. Northward to the equator, there is an eastward decrease of complexity which could be explained by the persistent cloudy days induced by the interaction flows northward Madagascar as monsoon and trade wind splitting interaction producing ITCZ (intertropical convergence zone). Moreover, complexity is higher over land (Horn of Africa) than over ocean as it is also seen in Figure 2 with coefficient of variation.

In Figure 8, we can notice that there is a striking similarity between in the latitudinal variation of complexity and the coefficient of variation. This means that AP method is capable to catch the variability of hourly solar radiation over the western part of the Indian Ocean. There is a good correlation between computation complexity of clear sky index and statistical description of hourly solar radiation variability. Even if the correlation is slightly lower for northern equatorial band, the coefficient of correlation remains high with a value around 0.7 for the southern equatorial and tropical bands. Thus, AP method and statistical variability are well positively correlated which means that computational complexity AP is in accordance with CV, which is a statistical measure of the solar radiation variability. The time variation of the hourly solar radiation can be tracked by the computational complexity AP method. As it is known, the annual and semiannual are the most dominant time mode of solar radiation on the earth. There is a latitudinal dependency of these two modes. The semiannual mode is most pronounced over the equatorial band whereas the annual mode is most pronounced over the tropical and extratropical bands. Both CV and AP method exhibit the annual mode south to 15°S (Figure 8a) and semiannual mode over the equatorial band (Figure 8b,c). For the former, it is in accordance with Bessafi et al. [9] for La Réunion (21°S/55°E) located in the south tropical band. He studied a different period (2011 to 2015) and showed that the complexity is roughly lower during austral summer due to the large-scale interannual perturbations prevailing over the Indian Ocean.

If we consider the algorithmic probability AP as a benchmark tool to compute the complexity of short strings, the KS-ES presented in this study shows also a significant ability to compute the complexity of hourly clear sky index in accordance with the AP method (Figure 9). We can notice a good correlation of roughly 0.7 for different tropical band for the 2004–2006 period and all the selected longitude (43°E, 52°E, 57°E, and 62°E).

### 4.2. Algorithmic Probability Method (AP) Versus Kolmogorov Complexity with Suggested Encoding Scheme (KC-ES)

KC-ES complexity is based on encoding the physical value of solar radiation into binary representation in base 2, which is completely different approach than the AP approach. Such interest on complexity approach for hourly solar radiation is mostly to focus on the predictability of the solar resource over our studied area. As mentioned previously, the cloud cover variability pattern is mostly driven by meteorological conditions prevailing over this tropical area. Thus, it is important to check how this new approach, to measure complexity with a new encoding process through a straightforward binarization for short strings, can give a realistic measure of complexity of the hourly solar radiation. The 2004–2006 period is time period in which we had a strong sea surface temperature anomaly over the south part of Mozambique Channel (SWIO SST), a strong MJO event in the Indian Ocean basin, and also tropical cyclones. We have selected these events to check how our new method is able to catch complexity for short strings deduced from hourly time samples in comparison to the AP method.

#### 4.2.1. South-West Indian Ocean SST (SWIO SST Index): Positive SST Anomalies during 2004–2006

As the Pacific Ocean, the Indian Ocean is an ocean which also experiences interannual oscillation with episodes of strong departure from its mean state. Roughly, there are drastic changes of the zonal circulation of the upper ocean due to surface stress wind by the lower level of the Walker cell. For the latter, the large scale of ascendant and subsident branch of the cell also move zonally. This air–sea interaction can affect in one place the tropical cyclone activity and cloud pattern variability during a long period around one or two years [19,23,24,37]. Such interannual oscillation over the Indian Ocean is not regular in time and can generate east–west dipole of sea temperature anomalies in the basin. Like El Nino, this phenomenon is commonly tracked with SST east–west dipole with positive-negative or negative-positive anomalies. In the literature, there are four indicators to state the sea surface anomalies in the Indian Ocean basin, which are the western tropical Indian Ocean (WTIO) SST index (50°E–70°E, 10°S–10°N), the southeastern tropical Indian Ocean (SETIO) SST index (90°E–110°E, 10°S–0°), the South-West Indian Ocean (SWIO) SST index (31°E–45°E, 32°S–25°S), and the dipole mode index (DMI) [24]. Each index corresponds to a temperature anomaly of the sea surface over a specified geographical area which mainly is located over the west (WTIO), east (SEITO) equatorial band, the South-West Indian Ocean (SWIO), and the over whole Indian Ocean (DMI). As mentioned previously, WTIO and SEITO SST anomalies form an east–west dipole. We build the dipole mode index (DMI) by computing the difference between WTIO and SETIO SST index. This index is also known as the Indian Niño index. Thus, DMI is positive when there is a warmer–colder dipole in respect to the east–west direction and conversely DMI is negative when there is a colder–warmer dipole. Then, there are irregular oscillations of SST anomalies in different region of the Indian Ocean which can be alternately warmer and colder through interannual and intraseasonal time scale. Moreover, all the SST time series index for the Indian Ocean covers the 1982–present time period and are freely available. During the 2004–2006 period, there were significant positive SST anomalies over the south of Mozambique Channel (SWIO SST index) which is depicted in Figure 10a. Significant means that the SST anomalies in absolute value are greater than the standard deviation indicated by the dashed line in the figure. Thus, we define three categories of the SWIO SST index state which are positive index for SST anomalies exceeding 0.5°C, neutral negative for SST anomalies between −0.5°C and 0.5°C and finally, negative index for SST anomalies less than −0.5°C. As an example, Figure 10b,c show the monthly mean SST anomaly over the south of Mozambique Channel for January 2005 (positive anomaly) and April 2005 (neutral anomaly which means that the absolute value of the SST anomaly is less than 0.5°C). The SWIO SST anomalies may influence climate over remote regions, cloud pattern variability, and then the solar radiation received on ground. Moreover, Figure 10c shows the regional pattern of SST anomalies during a neutral SWIO episode (April 2005) over the southern Mozambique Channel with a positive SST anomaly over the Mascarene area (La Réunion and Mauritius).

Figure 11 shows that the hourly clear sky index complexity during period time of positive SWIO SST index was highly variable within the south tropical band (30°–15°S) during the 2004–2006 period. In addition, same complexity pattern is highlighted by KC-ES complexity and AP method. The scale value of AP and KC-ES are two different measure of complexity. The former is obtained through numerical tables of probability distribution and the latter is a normalized value computed after the Lempel-Ziv process [8,9,10]. We can notice that over the Mozambique Chanel and Horns of Africa (longitude 43°E), the variability of complexity is highest for both AP and KC-ES methods than the remaining longitude transect (52°, 57°E, and 62°E). Moreover, the variability of complexity is on average of order twice higher for KC-ES complexity than the AP method. This means that KC-ES Kolmogorov complexity is more sensitive and capable to catch the day-to-day of hourly clear sky index variability and complexity.

#### 4.2.2. Madden–Julian Oscillation: Strong Event in April 2005

The Madden–Julian oscillation (MJO) is the largest atmospheric oscillation in the Indian Ocean basin. This oscillation was discovered in 1971 [26] and it is an eastward equatorial large-scale convective cell with a period of propagation between 30 days and 90 days [21,26,27,38]. This is an intraseasonal oscillation which has a global earth extend with an estimated eastward phase speed of 4.6 m/s. 

Figure 12a displays the phase diagram of the daily first two principal components PC1 (first principal component) and PC2 (second principal component) time series from multivariate EOF (empirical orthogonal function) of the MJO during January–June 2005 period. These two components were deduced from the real-time multivariate MJO (RMM) method which was extracted from the 850 hPa zonal wind, 200 hPa zonal wind, and OLR data [39]. For each day, the MJO amplitude and geographical location can be tracked in this phase diagram. The MJO is assigned to be in its neutral activity when its amplitude $\sqrt{PC1^{2}+PC2^{2}}$ is less than one standard deviation (indicated by a thick black circle in the figure). We can notice that the MJO strengthen over the Indian Ocean between 18 March 2005 and 2 April 2005 with a maximum over Australia (Maritime Continent) at 4 April 2005. The amplitude of the MJO over the western Indian Ocean is significantly high and exceeding twice the standard deviation of the MJO amplitude time series between 18 March 2005 and 20 March 2005 (Figure 12b).

Complexity in hourly clear index are computed with the AP method and KC-ES complexity for the eight phases of the MJO during the January–June 2005 time period. As shown in Figure 13, the results obtained with the KC-ES complexity are in a good agreement with those obtained with the AP method. The hourly clear index complexity is in average highest during phase 1 of the MJO at the outmost western longitude (43°S) where the MJO is active over Africa during this phase (Figure 12a). Moreover, the western Indian Ocean area between 52°E and 62E, exhibits higher complexity of clear sky index during phase 2 of MJO which is active in the Indian Ocean (Figure 12b–d).

#### 4.2.3. Tropical Cyclone: Austral Summer Season 2004–2005 and 2005–2006

The Indian Ocean is among the active tropical cyclone basins in the world [40]. Tropical activity in the South-West Indian Ocean is mostly important during austral summer season between November–April [41]. Few tropical cyclones are recorded in the northern basin in comparison to the southern Indian Ocean. Usually, tropical cyclogenesis is mainly located roughly out most of the 10°S–10°N latitude band. Among sixteen tropical cyclones which was recorded in the basin during 2004–2005 and 2005–2006 tropical cyclone season, there are nine tropical cyclones that was tracked in the western part of the southern Indian Ocean in the 30°S–15°S latitude (Table 1). Western part means here both the 52°E–62°E longitude and the Mozambique Channel.

The AP method and KC-ES complexity of clear sky during tropical episodes are displayed in Figure 14. As a reference, we have also plotted the complexity clear sky index for days without tropical cyclones in the selected area. For each method, we have on the left side, black dots which represent the distribution values of complexity for days without tropical cyclones. On the right side, we have plotted the complexity only for days where a tropical cyclone track crosses the 30°S–15°S/43°-62°E area during austral summer season 2004–2005 and 2005–2006. Despite the different scale, between AP and KC-ES methods, the variability of the complexity during tropical cyclone events is embedded inside the natural complexity of clear sky index prevailing during days without tropical cyclones events over the western part of the Indian Ocean. This is observed both with the AP method and KC-ES. This embedding distribution could be related with the lower infra-day variability of solar radiation of persistent cloudy days associated with tropical cyclones. Moreover, it means that convective systems prevailing over the western Indian Ocean induce high variability of solar radiation complexity. Tropical cyclones are ones among them which contribute to the variability of solar complexity embedded within a higher broadband range of complexity.

## 5. Conclusions

In this study, we propose a new simple encoding scheme to extend the application of Kolmogorov complexity to short time series for hourly solar radiation. To assess this new approach (KC-ES), we compared our results with the results obtained with AP method. The AP method is based on the coding theorem (CT) and it is very useful for computing the complexity of short string and used here as a benchmark. We analyzed the complexity of the hourly solar radiation time series from the satellite-derived HC3 Archives of Meteosat for the period February 2004–December 2006. The satellite observation covers 66°S–66°N/66°W-66°E with a resolution of about 8 km for the western part of the Indian Ocean which is our area of interest for solar studies [8,9]. We have extracted from this database, the solar radiation times series for 43°E, 52°E, 57°E, and 62°E longitude location for each one degree of latitude between 30°S and 15°N to study the performance of the KC-ES method over the area of the Indian Ocean covered by the satellite. Firstly, we showed the temporal and spatial variability of the solar radiation over our selected tropical area which is highly driven by tropical circulation and oscillation through seasonal, intraseasonal, daily, and infra-daily time scale. 

To overcome the influence of the natural daily and seasonal solar radiation inside a daytime over the tropics, we have used the clear sky index which is a good indicator of the atmospheric attenuation due to clouds which is known to have a direct impact on the solar radiation stochasticity. To test the ability of the KC-ES complexity method to compute complexity for short time series, we presented the encoding process done with the data. The day-to-day encoding solar radiation time series was obtained by converting each hourly solar radiation value by its binary representation in base 2. For each day, we converted a set of twelve hourly solar radiation values into a longer binary string which is the result of the concatenation of each individual binary string. Thus, the binarization of the hourly solar radiation is much more straightforward than the previous method done using a threshold method and the spectrum analysis to take account the sensitivity of the threshold method to compute complexity [8,9].

The coding theorem (CT) and the method to compute the complexity with AP method are also presented. Even if it is not necessary with the AP method, we have binarized the hourly time series by using the mean in the threshold method. Firstly, the results obtained with AP method show its ability to catch the variability and complexity of hourly solar radiation over the western part of the Indian Ocean. In addition, results obtained by the Kolmogorov complexity with suggested new encoding scheme (KC-ES) are in good agreement with those achieved by AP method. 

Following the global analysis, we performed tests to compute complexity with AP method and KC-ES complexity on three strong meteorological events which occur during the 2004–2006 period. For each case study, we presented (i) an important sequence of positive South-West Indian Ocean SST anomaly during the 2004–2006 period with a maximum during 2005, (ii) a strong MJO event also in 2005, and (iii) nine tropical cyclones which crossed the 30°S–15°S/43°-62°E area during the 2004–2005 and 2005–2006 austral summer. These case studies are performed to assess the ability of the Kolmogorov complexity with new encoding scheme (KC-ES) to catch the physical complexity for solar radiation over tropical region where the weather and cloud cover are mostly triggered by large-scale active convective systems through daily to interannual time scale. KC-ES complexity applied to the hourly solar radiation successfully capture the corresponding physical processes. This measure of complexity could also be used as an index in day-to-day hourly solar radiation classification.

## Figures and Tables

**Figure 1 entropy-21-00552-f001:**
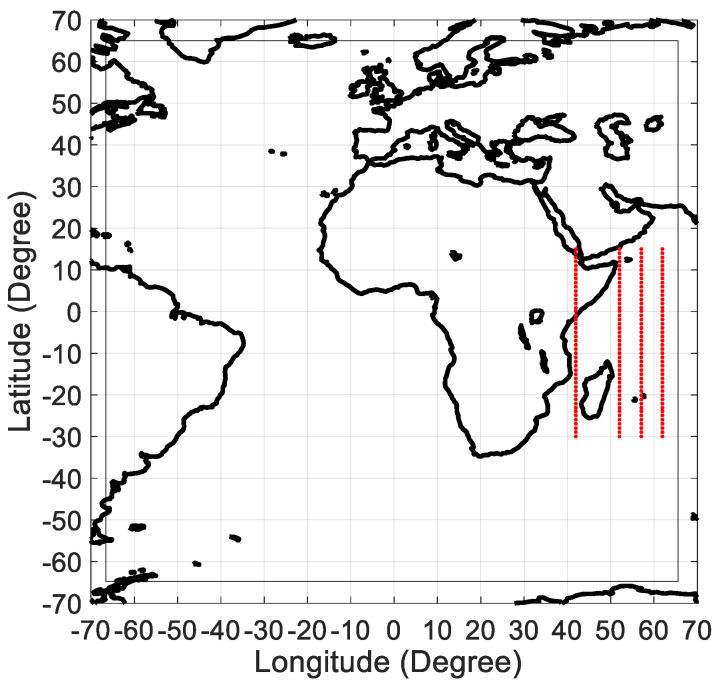
Spatial coverage of the satellite-derived HC3 archive (66°S–66°N/66°W–66°E). Four longitude transects (43°E, 52°E, 57°E, and 62°E) from 30°S to 15°N with a latitudinal increment which are highlighted in red dots.

**Figure 2 entropy-21-00552-f002:**
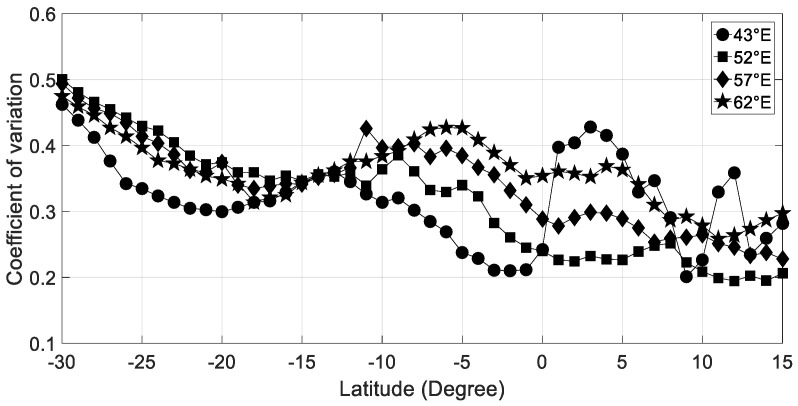
Latitudinal variability of the coefficient of variation of hourly solar radiation recorded during the 2004–2006 period for 43°E, 52°E, 57°E, and 62°E longitude transects.

**Figure 3 entropy-21-00552-f003:**
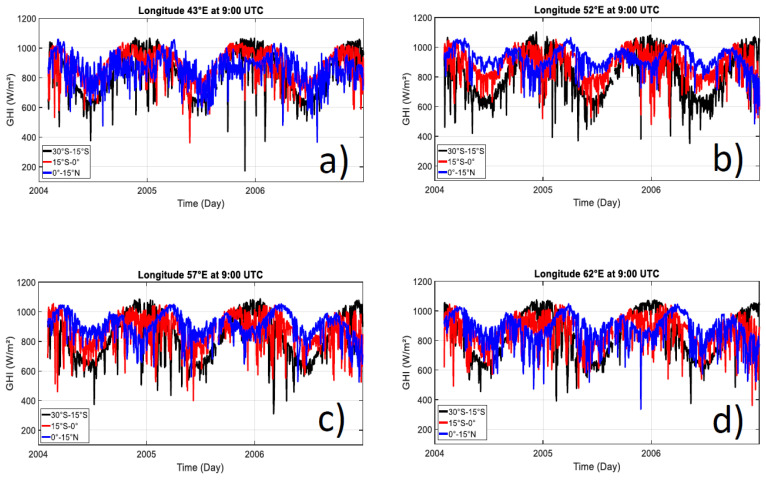
Latitudinal average band (30°S–15°S, 15°S–0°, and 0°–15N) of solar radiation time series at 9:00 UTC at longitude (**a**) 43°E longitude; (**b**) 52°E longitude; (**c**) 57°E longitude; and (**d**) 62°E longitude. GHI: Global horizontal irradiance.

**Figure 4 entropy-21-00552-f004:**
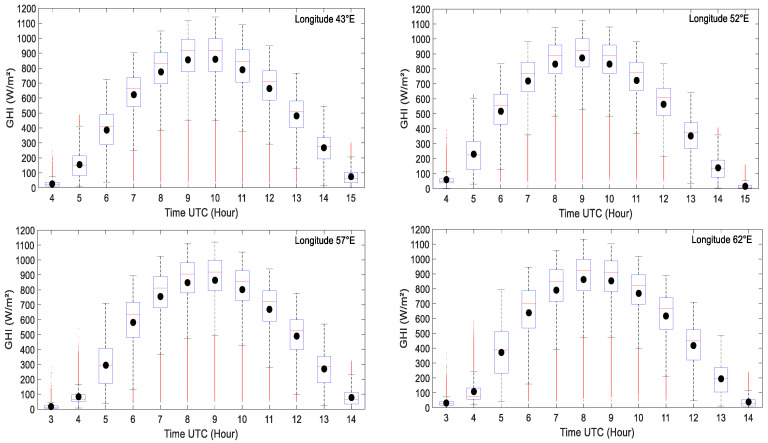
Boxplot of the hourly solar GHI (global horizontal irradiance) values recorded from 2 January 2004 to 31 December 2006 for 43°E, 52°E, 57°E, and 62°E longitude transects between 30°S and 15°N. Black dot represents the mean of solar radiation.

**Figure 5 entropy-21-00552-f005:**
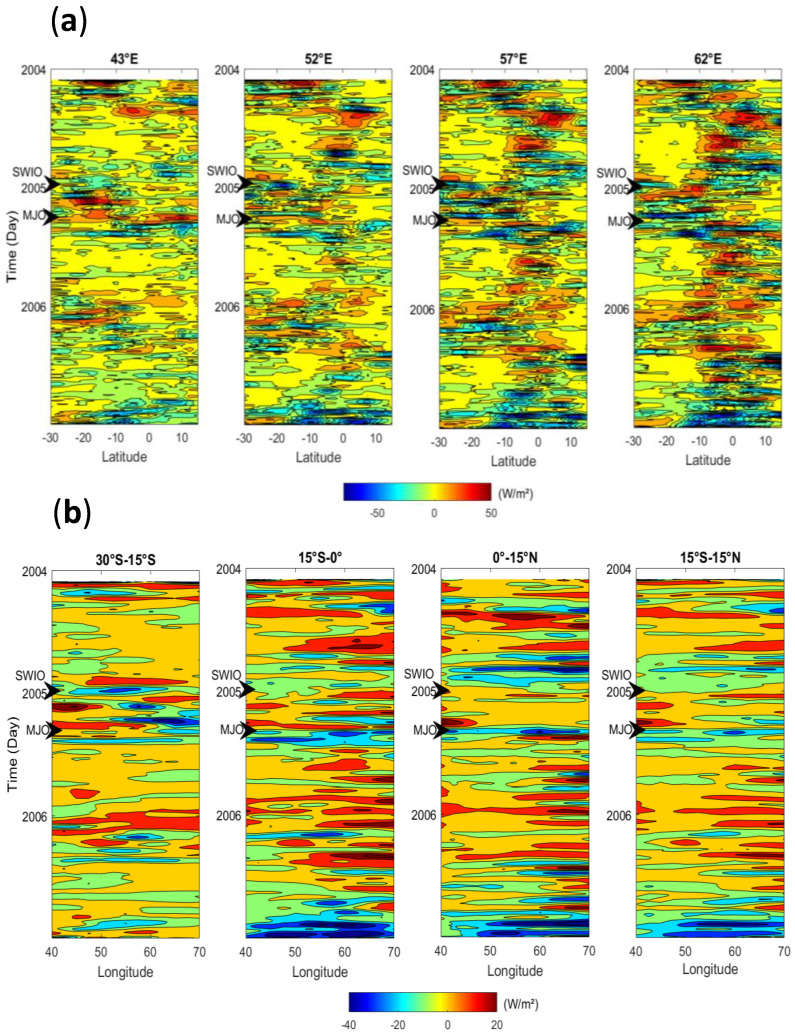
(**a**) Latitude-time ORL (outgoing longwave radiation) and (**b**) longitude-time OLR anomalies recorded during 2004–2006 period. Black arrows are indicative of the time when the 2005 Madden–Julian oscillation (MJO) event and SWIO (South-West Indian Ocean sea surface temperature (SST) index) were in their maximum intensity. The data is freely available (https://www.esrl.noaa.gov). Unit is W/m².

**Figure 6 entropy-21-00552-f006:**
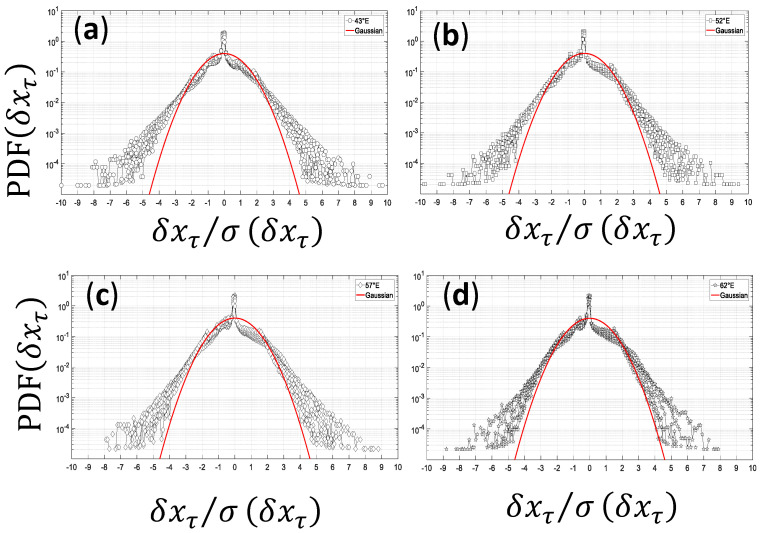
Semilogarithmic scale of normalized experimental PDFs (Probability Density Function) of the increment $\delta x_{\tau }$ of hourly clear sky index for time lag τ (τ =1, 2, 3, 4, 5, and 6 hours) for the period 2004–2006 record for (**a**) Longitude 43°E; (**b**) Longitude 52°E; (**c**) Longitude 57°E, and (**d**) Longitude 62°E and 30°S–15°N latitude band. Red line is the Gaussian PDFs.

**Figure 7 entropy-21-00552-f007:**
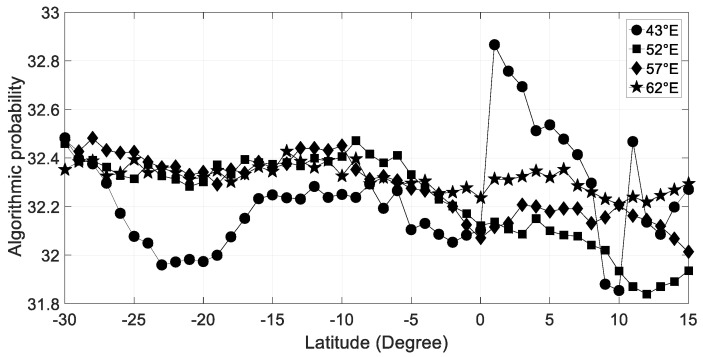
The time average over the 2004–2006 period of algorithmic probability (AP) method for hourly clear sky index versus latitude for 43°E, 52°E, 57°E, and 62°E longitude transects.

**Figure 8 entropy-21-00552-f008:**
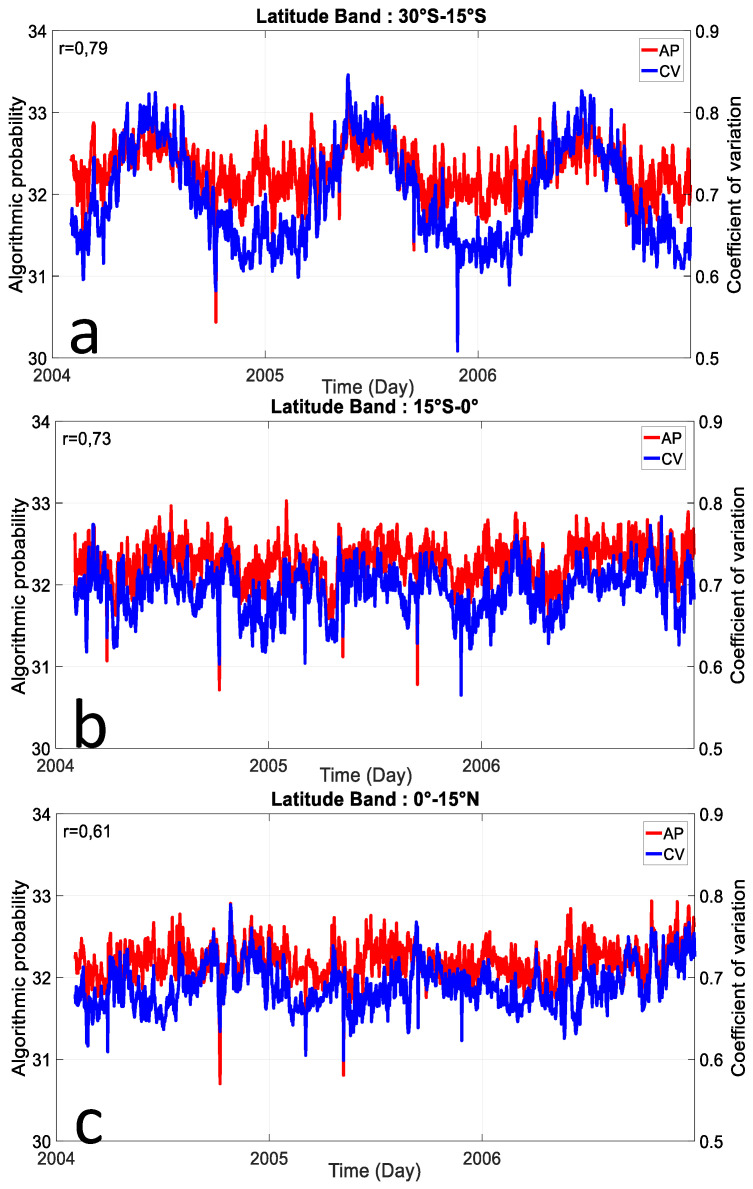
Longitude average (43°E to 62°E) of daily AP method of hourly clear sky index and coefficient of variation (CV) of hourly solar radiation during the 2004–2006 period for (**a**) 30°S–15°S, (**b**) 15°S–0°, and (**c**) 0°–15°N latitude band. The average is done for all longitude transects.

**Figure 9 entropy-21-00552-f009:**
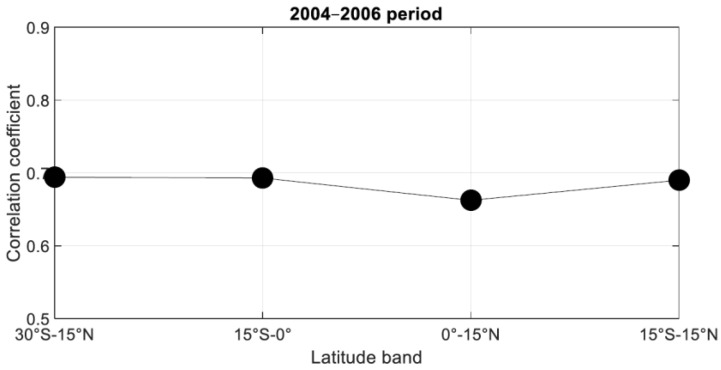
Correlation coefficient of longitude average (43°E to 62°E) of daily AP method and Kolmogorov complexity with suggested encoding scheme (KC-ES) complexity of hourly clear sky index during the 2004–2006 period for 30°S–15°S, 15°S–0°, and 0°–15°N latitude band. The average is done for all longitude transects.

**Figure 10 entropy-21-00552-f010:**
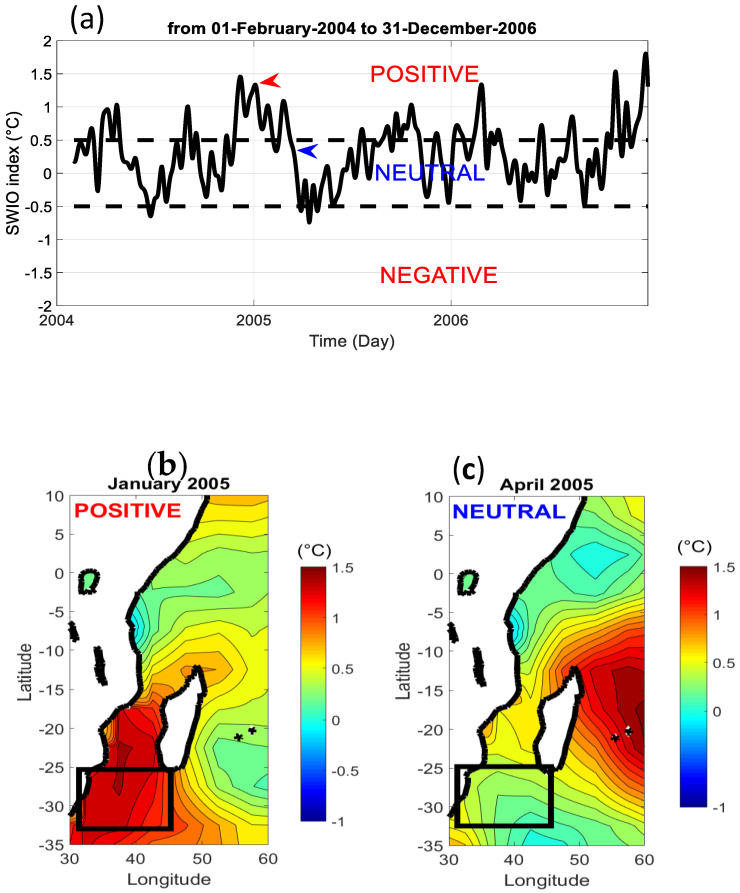
(**a**) SWIO SST index for 2004–2006 time period; (**b**) latitude–longitude monthly mean SST is displayed for January 2005 during a positive SWIO episode and (**c**) April 2005 during a neutral SWIO period. Black dashed lines correspond to one standard deviation while red and blue fields indicate the selected period for the map representation in (b) and (c). Black boxes indicate the area which is used for computing the SWIO SST anomaly. The weekly indices are freely available at (https://stateoftheocean.osmc.noaa.gov/sur/ind/) and the SST data is also freely available at https://www.esrl.noaa.gov/psd/.

**Figure 11 entropy-21-00552-f011:**
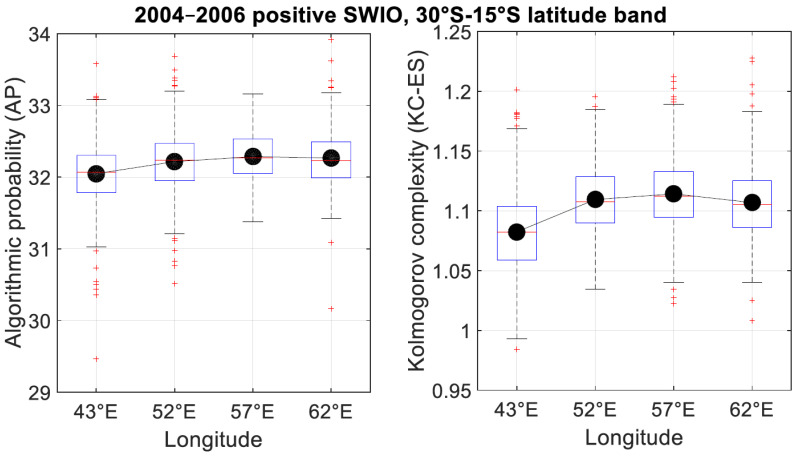
Boxplot of the hourly clear index complexity during positive SWIO episode (January 2005) between 30°S and 15°N for 43°E, 52°E, 57°E, and 62°E longitude transects for (**a**) AP method and (**b**) KC-ES complexity. Black dots represent the mean of the complexity.

**Figure 12 entropy-21-00552-f012:**
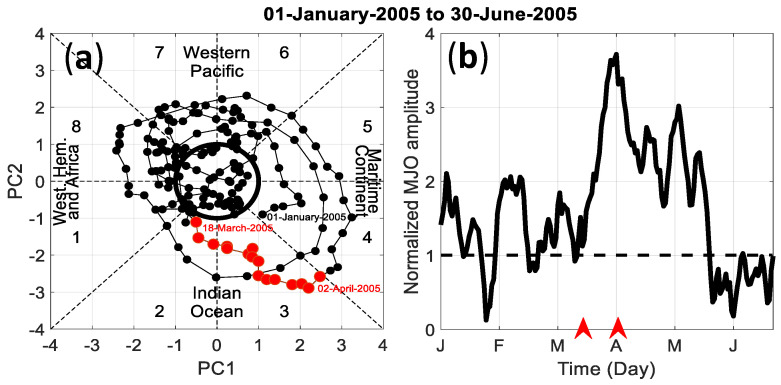
(**a**) PC1–PC2 (first principal component–second principal component) phase-space points of January–June MJO activity during 2005. Eight equal-angled phase-space categories are defined. For each category, the approximate locations of the enhanced convective signal of the MJO are also labeled. Thick black circle is the one standard deviation. Weak MJO is defined when its normalized amplitude is less than unity; (**b**) MJO amplitude time series from 1 January 2005 to 30 June 2005. Black tick dotted line is the one standard deviation. Red arrows are indicative of the time period of the MJO located over the Indian Ocean. The PC’s and the phase of the real-time multivariate MJO (RMM) time series are available online (http://www.bom.gov.au/climate/mjo/).

**Figure 13 entropy-21-00552-f013:**
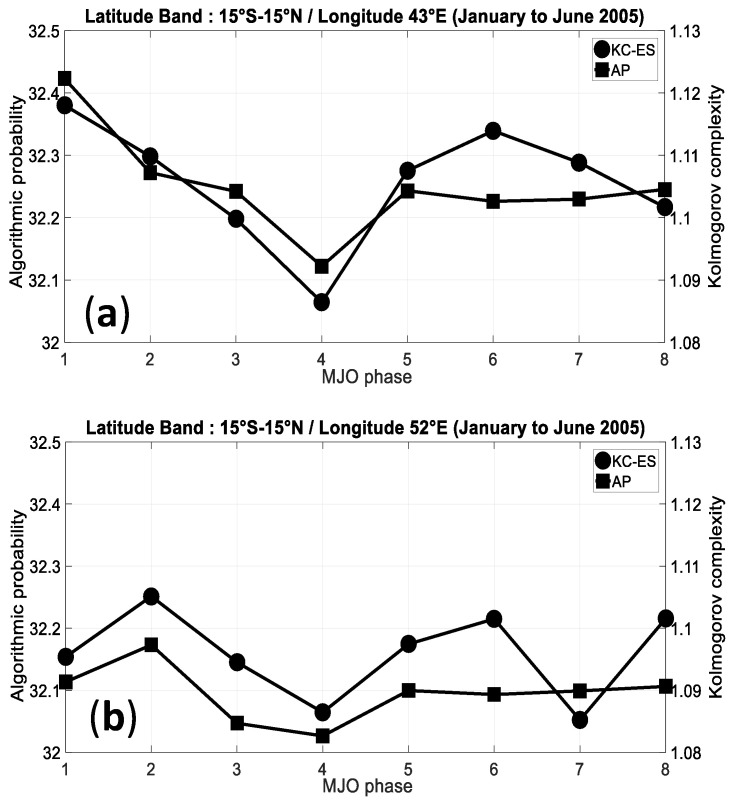
Average of AP and the KC-ES complexity of the clear sky index during the January–June 2005 MJO activity for different phases of the MJO within the 15°S–15°N latitude band for (**a**) the longitude 43°E; (**b**) the longitude 52°E; (**c**) the longitude 57°E; and (**d**) the longitude 62°E.

**Figure 14 entropy-21-00552-f014:**
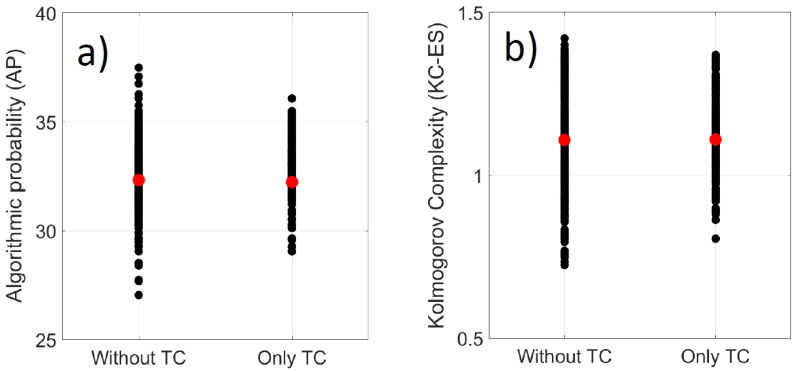
Distribution of complexity of clear sky index computed for two meteorological cases. (without and only tropical cyclone (TC)) with (**a**) AP method and (**b**) KC-ES complexity. Thick red dots indicate the mean of the distribution.

**Table 1 entropy-21-00552-t001:** List of the tropical cyclone which crossed the 30°S–15°S/43°E–62°E area during the austral summer of 2004–2005 and 2005–2006.

43°E–62°E Longitude Band & 30°S–15°S Latitude Band			
Tropical Cyclone Name	Cyclogenesis Date	Life of Time Days	Crossing Time Days
BENTO	19 November 2004	16	2
ERNEST	16 January 2005	10	3
DAREN	17 January 2005	8	2
FELAPI	26 January 2005	9	3
GERARD	29 January 2005	9	3
BOLOETSE	20 January 2006	18	11
920052006	18 January 2006	6	6
CARINA	22 January 2006	18	2
DIWA	2 March 2006	10	7

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
