# Peer review of "Algorithmic Probability Method Versus Kolmogorov Complexity with No-Threshold Encoding Scheme for Short Time Series: An Analysis of Day-To-Day Hourly Solar Radiation Time Series over Tropical Western Indian Ocean"

_entropy, 2019, doi:10.3390/e21060552_

Round 1

Reviewer 1 Report

I am not competent for the climate issue. My opinion only concerns issues related to methods

The work has the great interest to show that the two methods used give compatible results, which is a way to validate both methods.

It is important to have two compatible schemes to ensure robust results. The search for such schemes will undoubtedly continue. Other variants are possible. I think we should probably prefer those based on AP who have a local ability to see what happens. Those based on KC and its variants do not really seem likely to give better and are more likely to produce artifacts.

A new area of research comes from this type of application. The work has a practical interest. It also has a theoretical interest since it validates distant propositions of complexity measures which are theoretically linked, which the results confirm.

Line 98 

"However, the binarization process is still needed even for short solar radiation time series.ʉ۬"

No it is no true. The use of the coding theorem do not necessitate binarization. 

The following paper give some examples of the use of the coding theorem with non binary sequence.

Gauvrit, Nicolas, et al. "Algorithmic complexity for psychology: a user-friendly implementation of the coding theorem method." Behavior research methods 48.1 (2016): 314-329.

Behav Res (2016) 48:314–329   DOI 10.3758/s13428-015-0574-3 

https://link.springer.com/article/10.3758/s13428-015-0574-3

Ligne 569

"It is pointed out that AP method needs binarizing the hourly time series which is done using the mean in the threshold method." 

Idem

Author Response

Dear Reviewer 1,

Please fin as attached file our response to your fruitfull comments about our paper.

Best regards

Pr. M. BESSAFI

Reviewer 2 Report

In this paper the authors present a new approach based on compression to analise the potential risks to solar produced energy. This paper is a subsequent paper of part of the authors ([8] and [9] in the references) where the authors present new ways to overcome the difficulty of dealing with short time series.

In section 2, mainly regarding Algorithmic information, Kolmogorov complexity and the encoding I have a few comments to improve the manuscript. The major drawback of Kolmogorov complexity is not being computable in the sense that it is not possible for a given string and a fixed Turing machine to compute its value. This is why it is inaccurate and depends largely on the compressor that one uses to approximate it. Furthermore, since we are dealing with TM, in line 139 the expression "not easy to state" should be replaced by "impossible to state" as knowing whether a program stops for a given input is uncomputable. How exactly is the approximation of  m do you consider? This needs to be properly explained for sake of clarification of the reader. Just saying that m can me approximated by any probability distribution is not enough. It must be computable and chose properly to feat the data.  In subsection Kolmogorov complexity I think the order of the operations is reversed, first you apply the threshold and obtain a binary sequence which then is compressed with LZ. The introduction to normalized compression should be explained in the context of the paper, specially if you are using short binary strings. Please correct equation (1). The encoding presented in line 179 looks strange for me specially if you are concatenating the strings. Notice that if you use different size for representing the values in binary how would you recover the initial values from the enconding?

In my opinion, section 3 is too extensive and should focus on the main aspects that are relevant for the analysis. For example it can be explained succinctly that the data used is related with solar exposition and there are climate factor that influence that value. The full details of it affects could be omitted for the sake of understanding and readability of the paper. 

The new scheme presented by the authors to analyse the short time series is hard to identify in the paper and therefore it is difficult to understand the main differences from this approach to the previous one [8] and [9]. 

In Section 4 the figures are too big and should be rescaled. Furthermore, there is a repetition of the text to describe again the climate influences on light exposure. I have a comment on line 399 and the respective figure. In Figure 6 the red line does not represent a Gaussian distribution as it does not have tail values. This must be revised and presented properly. Also, the importance of this picture as well as Figure 7 for the presentation of the results must be clear for the reader. Caption of figure 6 cut in two pages is terrible. The reader cannot distinguish caption from text. 

Many claims are not proven, calculated or are given any insight of their validity. For example:

1) In line 362 the authors claim that the decrease of complexity is explained by more complex flows. How can this be? If the flows are more complex one would expect that their description of their effect to be increasing the complexity. This also need more explanation. 

2) Line 377. "Thus, AP method and statistical variability are well positively correlated which means that computational complexity AP is in accordance with physical variability." Why is that?

Furthermore, how can you deduce that the results observed can be explained by the semi-anual cycle? Why is that?

Analysing figure 8 it seems to me that AP and CV are correlated but this is some evidence that is already presented in the previous works. Why this is presented here? The focus of the paper should be comparing the new approach to this one, not giving evidences that AP is good to characterise CV.

In section 4.2.1 the authors compare AP with KC-ES and claim that they have similar shape (graphically visible) but the variability is twice for KC-ES than for AP, but this is not clear from the text nor the graphic. Furthermore the scale of the two graphics in Fig 11 are different (~30 for AP and ~1 for KC-ES). The same issue arises in Figure 14. I do not see why the phenomena is important in the comparison between the two methods.

In the conclusions the authors the authors suggest why the analysis was performed in the strong meteorological evens but I do not understand why this useful. I would expect to use this approach to identify possible occurrences of this phenomena or to predict day light exposure.

Summing up, the paper is to extensive to support the idea of how to use KC meteorological data. I do not fully understand the goal of the paper and why in fact this new approach is better than the ones presented in [8] and [9]. Short-time series is indeed a good motivation but in my opinion the paper fails to properly present that motivation and why this is really useful in this context.

Author Response

Dear Reviewer 2

Please find our response to your fruitfull comments about our paper.

Best regards

Pr. M. BESSAFI

Round 2

Reviewer 2 Report

In my opinion the responses of the authors to my comments are satisfactory but they fail reflect these effort in the paper. For example:

1) No improvement in the presentation of the differences for the other 2 papers is made.

2) The presentation of the encoding is not justified. Notice that to avoid to have the length explicitly written is to pad the encoding with 0's to the left to a fixed size. I believe that such encoding is more efficient for compression than the one presented by the authors.

3) They do not clearly highlight the new scheme used that is different from the other 2 papers already published. 

4) If,  in fact, AP was never used previously this should be mentioned in the abstract and in the paper. Furthermore, if it was never used in this context how can this be used as a benchmark and why being comparable to KC-ES makes this approach useful.

5) They do not correct de definition of S(x) in eq. 1.

6) Figure 6 there is no Gaussian distribution represented there. This must be corrected.

7) The idiomatic expressions and the entire english of the manuscript should be revised. 

The other issues that I raised previously I agree that is a matter of opinion and respect the authors choice not to follow my suggestions. I think that in order to make the paper clear for the reader the authors have to do the effort to write in the paper the relevant aspects mentioned above and that they put in the reply to my comments. Therefore, I keep my first recommendation for this paper.

Author Response

Dear Reviewer 2 (round2)

Please find as attached file our response to your comments of our paper entitled " Algorithmic probability method versus Kolmogorov complexity with no-threshold encoding scheme for short time series: an analysis of day-to-day hourly solar radiation time series over tropical Western Indian Ocean".

Regards

Pr. M. BESSAFI

Round 3

Reviewer 2 Report

Answering directly to the authors:

"The fact is that the reviewer is always in a situation to go up against the paper by finding the weak spots of it in the function of improving the text and readability of the paper."

This is not the case as I had have mentioned: "The answers  given were satisfactory", but it is a fact that the authors did not put the required effort to improve the paper. They only reply not correct the paper significantly.

"Namely, there is no possibility (and it is not necessary) to compare the presentations and results in papers Mihailović et al. (2018) – (P1), Bessafi et al. (2018) – (P2) and paper by Bessafi et al. (2019) – (P3), which is under the reviewing process." 

You are wrong. This is what is called "literature reviewing". This is a crucial part in the paper specially when there are two quite similar papers in the same topic that, if do not state it properly may look like "milking the cow" with another another set of data. Instead of  La Reunion it considers a bit wider area of same region.

Answering to authors reply to my previous concern number 3: I stand with my comment. Maybe the problem is the Lempel-Ziv compressor that is not suitable to lead with this type of strings. You should try other compressor. 

Regarding the answer to comment 4: The results emphasise something that it is already well known and the that authors also acknowledge in section  Coding theorem and Algorithmic probability method. 

Answer to 5: Now it is in standard mathematical notion. When presenting notation and results this must be carefully treated as it might lead to confusions to the reader. 

Answer to 6:  How can an "inverse of a parabola" be a Gaussian PDF? A Gaussian is defined using an exponential function. This must be removed or restated otherwise is completely wrong.

Author Response

Dear Reviewer,

Thank you for your comments. Please find our point-by-point answer as attached file.

Best regards

Prof. M. BESSAFI
